# The Relationship between Ectomycorrhizal Fungi, Nitrogen Deposition, and Pinus massoniana Seedling Nitrogen Transporter Gene Expression and Nitrogen Uptake Kinetics

**DOI:** 10.3390/jof9010065

**Published:** 2022-12-31

**Authors:** Pengfei Sun, Ruimei Cheng, Wenfa Xiao, Lixiong Zeng, Yafei Shen, Lijun Wang, Tian Chen, Meng Zhang

**Affiliations:** 1Ecology and Nature Conservation Institute, Chinese Academy of Forestry, Key Laboratory of Forest Ecology and Environment of National Forestry and Grassland Administration, Beijing 100091, China; 2Co-Innovation Center for Sustainable Forestry in Southern China, Nanjing Forestry University, Nanjing 210037, China

**Keywords:** nitrogen addition, ectomycorrhiza fungi, *P. massoniana*, NH_4_^+^ and NO_3_^−^ uptake kinetics, AMT family gene, NRT family gene

## Abstract

Analyzing the molecular and physiological processes that govern the uptake and transport of nitrogen (N) in plants is central to efforts to fully understand the optimization of plant N use and the changes in the N-use efficiency in relation to changes in atmospheric N deposition changes. Here, a field experiment was conducted using the ectomycorrhizal fungi (EMF), *Pisolithus tinctorius* (Pt) and *Suillus grevillei* (Sg). The effects of N deposition were investigated using concentrations of 0 kg·N·hm^−2^a^−1^ (N0), a normal N deposition of 30 kg·N·hm^−2^a^−1^ (N30), a moderate N deposition of 60 kg·N·hm^−2^a^−1^ (N60), and a severe N deposition of 90 kg·N·hm^−2^a^−1^ (N90), with the goal of examining how these factors impacted root activity, root absorbing area, NH_4_^+^ and NO_3_^−^ uptake kinetics, and the expression of ammonium and nitrate transporter genes in *Pinus massoniana* seedlings under different levels of N deposition. These data revealed that EMF inoculation led to increased root dry weight, activity, and absorbing area. The NH_4_^+^ and NO_3_^−^ uptake kinetics in seedlings conformed to the Michaelis–Menten equation, and uptake rates declined with increasing levels of N addition, with NH_4_^+^ uptake rates remaining higher than NO_3_^−^ uptake rates for all tested concentrations. EMF inoculation was associated with higher V_max_ values than were observed for non-mycorrhizal plants. Nitrogen addition resulted in the upregulation of genes in the AMT1 family and the downregulation of genes in the NRT family. EMF inoculation under the N60 and N90 treatment conditions resulted in the increased expression of each of both these gene families. NH_4_^+^ and NO_3_^−^ uptake kinetics were also positively correlated with associated transporter gene expression in *P. massoniana* roots. Together, these data offer a theoretical foundation for EMF inoculation under conditions of increased N deposition associated with climate change in an effort to improve N absorption and transport rates through the regulation of key nitrogen transporter genes, thereby enhancing N utilization efficiency and promoting plant growth. Synopsis: EMF could enhance the efficiency of N utilization and promote the growth of *Pinus massoniana* under conditions of increased N deposition.

## 1. Introduction

Atmospheric nitrogen (N) deposition is increasing due to the excessive emission of active N compounds produced by anthropogenic activities, and its impact on ecosystems has become a hot topic in global environmental quality and climate change [1,2]. As the largest developing country, China is one of the three hot spots of high N deposition worldwide. Although the earlier rapid increases in N deposition have stabilized, the total amount remains at a relatively high level [3]. Atmospheric N deposition can influence plant growth, absorption, and nutrient distribution through both direct and indirect mechanisms [4]. N is essential for plant growth and development, and both environmental and soil N levels can influence plant N assimilation and transport [5,6,7]. N is present in soil in the forms of organic N and inorganic N, including nitrate ions (NO_3_^−^) and ammonium ions (NH_4_^+^). Of these, only inorganic N is thought to be important for plant growth [8,9,10].

A comprehensive overview of the plant N cycle and its association with the surrounding ecosystem necessitates insight into the N preferences of a given plant species, available N pools, and associated N flux [11]. It is thus important that the kinetics of N uptake by plant roots and associated transporter gene expression dynamics be examined. Plant nutrient uptake can be effectively summarized by the V_max_ and K_m_ parameters, with the former denoting the maximum rate that a root system can absorb external ions and the latter corresponding to the affinity of plants for particular ions. Larger V_max_ values are indicative of a greater potential for a given plant to absorb a given ion, while a smaller K_m_ value is indicative of greater plant affinity for these ions. The α value additionally reflects the rate of nutrient flow into the root system, with higher α values denoting more rapid nutrient ion flow [12]. Uptake kinetics are widely studied when examining plant nutrient absorption characteristics. N addition can reduce the N demands of *Populus tremuloides*, contributing to a reduced V_max_ corresponding to the uptake of NH_4_^+^ and NO_3_^−^ by these plants [13]. Analyses of potatoes exhibiting varying levels of N-use efficiency have shown that potatoes with higher N-use efficiency show larger V_max_ values and smaller K_m_ values corresponding to NH_4_^+^ and NO_3_^−^ [14]. In kale, absorption analyses have revealed lower levels of affinity for NH_4_^+^ and corresponding V_max_ values compared with those for NO_3_^−^ [15].

NH_4_^+^ and NO_3_^−^ absorption in plants is primarily an active process mediated by ammonium ion and nitrate transporters that govern both the uptake and transport of these nutrients [10,16]. The AMT1 and AMT2 gene families include most of the transporters responsible for NH_4_^+^ absorption and transport in plants, with one of these families encoding the low-affinity transport systems (LATS) that are important when NH_4_^+^ concentrations range from 500 µmol/L to 50 mmol/L and the other encoding the high-affinity transport systems (HATS) that are important when NH_4_^+^ concentrations range from 2.5 to 350 µmol/L [17]. Proteins of the nitrate transporter (NRT) family control NO_3_^−^ uptake, and this family includes both LATS components that are active when the K_m_ value exceeds 0.25 mmol/L and HATS components that are dominant at K_m_ values of less than 6–100 µmol/L [18]. Feedback mechanisms mediated through the differential utilization of these LATS and HATS systems can control endogenous N assimilation in response to exogenous N availability, allowing plants to more effectively utilize N through the appropriate up- or downregulation of these different gene systems as needed. In citrus, NH_4_^+^ levels were found to regulate NH_4_^+^ HATS activity and CitAMT1 expression, both of which were downregulated at high N levels [19].

Ectomycorrhizal fungi (EMF) are symbiotic with many plants. This mycorrhizal association allows young plant roots to be closely wrapped by fungal mycelia in order to grow hyphal sheaths, which can continue to grow mycorrhizal hyphae instead of root hairs and extend extraradical mycelia into the soil [20]. EMF are capable of promoting host plant growth and nutrient absorption, thus improving plant resistance to biotic and abiotic stressors [21,22]. A growing number of studies have examined the role that EMF play in transforming soil N-containing compounds and regulating plant N utilization [23,24]. Soil N availability can profoundly impact the plant N acquisition dynamics and N source preferences of a plant. EMF species such as saprotrophs can produce oxidative and hydrolytic exoenzymes, utilizing organic soil N sources including free amino acids and high-molecular-weight N-containing compounds such as proteins, polyphenol–protein complexes, and chitin, which are hydrolyzed to produce NH_4_^+^ or amino acids that can then be transported into plant root systems [25]. As such, under conditions of low NH_4_^+^ and NO_3_^−^ availability, EMF can enable plants to utilize otherwise inaccessible N sources. EMF inoculation has been shown to enhance *Populus tremula* × *tremuloide* ammonia N absorption, with the concomitant upregulation of three AMT genes [26].

China has the largest number of plantations in the world. Afforestation is crucial for regulating climate, reducing carbon dioxide emissions [27], and achieving the goals of carbon peak by 2030 and carbon neutralization by 2060 in China [28]. *Pinus massoniana* (Lamb.), which is a pioneer tree species of afforestation in Southern China, is also a typical ectomycorrhizal tree species. Research on the symbiotic relationship between *P. massoniana* and ectomycorrhiza has been conducted for many years. As early as 1989, Chen investigated and identified the symbiotic mycorrhiza of *P. massoniana* and found that a total of 27 species of EMF could form a symbiosis with it [29]. Among them, *Suillus grevillei* (Sg) and *Pisolithus tinctorius* (Pt) are typically excellent symbionts agents of *P. massoniana* [30].

Prior studies investigating the effects of EMF inoculation on *P. massoniana* have largely focused on growth or photosynthetic activity, and there has been little research exploring the uptake kinetics for NH_4_^+^ and NO_3_^−^ under conditions of N deposition [31,32]. This study was thus designed to investigate the effects of EMF inoculation in the presence of a range of N concentrations on the N absorption kinetics of *P. massoniana* and the expression of ammonium and nitrate transporter genes. This study further sought to elucidate the mechanisms underlying N uptake and utilization, with a particular focus on (i) whether EMF inoculation can alter the kinetics of N uptake or related transporter gene expression and (ii) whether the V_max_ values for NH_4_^+^ and NO_3_^−^ decline with increasing levels of N addition. It was hypothesized that excessive N deposition may result in an imbalanced reaction at higher levels of nitrogen addition, while EMF inoculation may be sufficient to regulate LATS- and HATS-related gene expression in a manner conducive to improved N uptake.

## 2. Materials and Methods

### 2.1. Research Site

Experiments were conducted at the Forest Ecosystem State Positioning Observation Station in the Three Gorges Reservoir area (110°540 E, 30°530 N, 375 m altitude), Zigui County, Hubei Province, China. *P. massoniana* is one of the main coniferous species in this area, and the proportion of total area and total stock is as high as 48.8% and 64.2%, respectively [33]. Additionally, the nitrogen deposition is 30 kg·N·ha^−1^a^−1^ [34].

### 2.2. Ectomycorrhizal fungi and Plant Materials

We designed a two-factor randomized block treatment. The first factor was EMF, which were *Suillus grevillei* (Sg), *Pisolithus tinctorius* (Pt) and CK (without EMF inoculation); EMF were provided by the Institute of Forest Ecological Environment and Nature Conservation, Chinese Academy of Forestry Sciences. The number of Sg was 864,647, and the number of Pt was 871,973. The second factor was N addition treatment, which was divided into four levels according to the annual atmospheric N deposition in this area: 0 kg·N·ha^−1^a^−1^, 30 kg·N·ha^−1^a^−1^ (normal deposition), 60 kg·N·ha^−1^a^−1^ (moderate deposition), and 90 kg·N·ha^−1^a^−1^ (excessive deposition). There were a total of 12 treatments, and each treatment had 100 pots, with a total of 1200 pots. Before the N addition, the seedlings were grown for three months to establish the symbiotic relationship between the seedlings and EMF. N was then added once a month in the form of NH_4_NO_3_ solutions containing 0, 0.714, 1.428, and 2.143 g/L of NH_4_NO_3_.

In March 2021, 1200 one-year-old *P. massoniana* seedlings were randomly selected and grown in a greenhouse, with each seedling planted in a separate pot (20 cm in diameter and 15 cm in height) filled with 3 kg of soil. The soil was collected from the *P. massoniana* forest stands within 2 km of the experimental site and was sterilized with high-temperature sterilization. The basic physical and chemical properties of the experimental soil were as follows: total soil N was 1.01 g·kg^−1^, total soil P was 0.54 g·kg^−1^, total soil K was 1.53 g·kg^−1^, available soil N was 47.28 mg·kg^−1^, available soil P was 8.97 mg·kg^−1^, available soil K was 90.17 mg·kg^−1^, organic matter content was 13.97 g·kg^−1^, and pH was 5.97. Three months after inoculation, 30 seedlings were randomly selected from each treatment. The infection rate of EMF was observed with a microscope. It was found that all the plants inoculated with EMF had been infected (Figure 1).

### 2.3. Root Absorbing Area

The root volumes of the *P. massoniana* seedlings were assessed using a drainage approach. Briefly, a methylene blue solution (0.064 mg/mL) was poured into three numbered beakers, each of which contained a volume 10 times the root volume. The seedlings were immersed in each beaker for 90 s, after which a 1 mL sample was collected from each beaker and diluted 10-fold. The absorbance was then measured at 660 nm using an ultraviolet spectrometer (UV mini 1240, Shimadzu, Kyoto, Japan), and the root absorbing area was calculated as follows:
Total absorbing area (m^2^) = [(C − C_1_) × V_1_] + [(C − C_2_) × V_2_] × 1.1


Active absorbing area (m^2^) = [(C − C_3_) × V_3_] × 1.1

where C (mg mL^−1^) is the initial concentration of the methylene blue solution; C_1_, C_2_, and C_3_ (mg mL^−1^) are the concentrations of the methylene blue solution after three root immersions; and V_1_, V_2_, and V_3_ (mL) are the volumes of the methylene blue solution after three root immersions.

### 2.4. Nitrogen Uptake Kinetics

A conventional depletion approach was used to assess the kinetics of seedling nitrate and ammonium absorption. Briefly, seedlings from appropriate treatment groups were collected and rinsed, and roots were dried with paper prior to transplantation into a container containing 500 mL of CaSO_4_ (2 mM) for 4 h. This was followed by transfer into a solution in which NH_4_Cl was the only NH_4_^+^ nitrogen source and KNO_3_ was the only NO_3_^−^ nitrogen source, with concentrations of 0.01, 0.05, 0.1, 0.1, 0.5, or 1 mM. Following a 6 h absorption period, 2 mL of the absorption solution was collected for analysis with the Smartchem200 (Smartchem200, ALLIANCE, BRESCIA, Italy) instrument, and NH_4_^+^ and NO_3_^−^ absorption rates were computed.

V (μmol g^−1^h^−1^) = (c_1_ − c_2_)/(t × m)

where c_1_ is the initial concentration of NH_4_^+^ and NO_3_^−^ (µmol mL^−1^), c_2_ is the final concentration of NH_4_^+^ and NO_3_^−^ (µmol mL^−1^), t is the absorption time (h), and m is the root dry weight (g).

Ion absorption curves were constructed in Origin 2021 (Origin Lab Corp., Northampton, MA, USA) based on the initial concentrations and absorption rate values computed for the absorption solution using the Michaelis–Menten kinetic equation, yielding the K_m_ and V_max_ values for NH_4_^+^ and NO_3_^−^ uptake.

### 2.5. qPCR

Harvested root and leaf samples were snap-frozen in liquid nitrogen prior to storage at −80°C. Total RNA was extracted from the seeding roots with a commercial kit (Aidlab Biotechnologies Co., Ltd. Beijing, China), and RNA purity and concentration were analyzed using an ND−2000 spectrometer (Thermo Fisher Scientific, Wilmington, DE, USA). RNA integrity was analyzed with 1.5% agarose gel electrophoresis (180 V, 15 min). A Takara PrimeScript RT Master Mix was used for cDNA synthesis, and qPCR reactions were performed using a 7300 Real-Time PCR instrument (Applied Biosystems, Carlsbad, CA, USA) with the primers listed in Table 1.

### 2.6. Data Analysis

Statistical analysis was performed using the SPSS 17.0 software package (SPSS, Chicago, IL, USA). A two-factor analysis of variance (ANOVA) followed by Duncan’s multiple comparisons test was used. The data shown are the mean and standard error (SE). A two-way-analysis-of-variance (ANOVA) was used to test the effects of the different labeling times and sampling times on the variables under evaluation. All figures were drawn using Origin software 2021 (Origin Lab Corp., Northampton, MA, USA).

## 3. Results

### 3.1. Root Activity and Absorbing Area

Nitrogen addition resulted in significant increases in the *P. massoniana* seedling root dry weight, root activity, root active absorbing area, and total root absorbing area. Specifically, these parameters initially rose and then declined with rising N concentrations, peaking at N60 before declining at N90. At each individual N concentration level, plants that underwent EMF inoculation exhibited significant increases in root dry weight, root activity, root active absorbing area, and total root absorbing area. At N60, the root dry weights of the plants inoculated with Sg and Pt rose by 182.97% and 211.18%, respectively, with corresponding increases of 66.78% and 81.64% in root activity, increases of 140.18% and 147.38% in the active absorbing area, and increases of 124.76% and 136.36% in the total absorbing area (Table 2). These data indicated that N addition, mycorrhizal symbiosis, and interactions between the two all increased the ability of the underground root system to absorb nutrients. The two-way ANOVAs further confirmed that mycorrhizal symbiosis significantly affected root dry weight and active absorbing area (*p* < 0.01) (Table 3).

### 3.2. NH_4_^+^ Uptake Kinetics

Increased NH_4_^+^ uptake rates were observed in response to EMF, N addition, and the interaction between N addition and EMF inoculation in accordance with the Michaelis–Menten equation. The rate of NH_4_^+^ uptake sharply increased under the N0 conditions in the presence or absence of EMF inoculation, whereas these upward trends were slower with lower levels of N addition. Uptake only reached saturation under the CK+N90 conditions (Figure 2). The V_max_, K_m_, and α values for the *P. massoniana* seedling NH_4_^+^ uptake rates derived from the Hofstee conversion formula showed that the V_max_ and α values declined with increasing levels of N addition but were significantly higher at any given level of N addition when EMF were also added relative to the CK conditions. The K_m_ for the N60+Pt conditions was lower than for all other conditions, indicating that the roots of the plants under these conditions exhibited the highest affinity for NH_4_^+^, such that plants were more effectively able to absorb NH_4_^+^ (Table 4).

### 3.3. NO_3_^−^ Uptake Kinetics

The NO_3_^−^ uptake rates were also found to follow the Michaelis–Menten equation under all tested treatment conditions. Specifically, these NO_3_^−^ uptake rates sharply rose in the 0.01–0.5 mM concentration range but reached saturation at concentrations from 0.5 to 1.0 mM. When the ammonium N solution concentration was the same as that of the nitrate N solution, the NH_4_^+^ uptake rate remained higher than the NO_3_^−^ uptake rate (Figure 2 and Figure 3). The V_max_ and α values for seedling NO_3_^−^ absorption rose with increasing N addition, and both of these values were higher for plants inoculated with EMF relative to plants under the CK conditions at any given N application level. K_m_ levels trended downwards with increasing levels of N addition, suggesting that high N concentrations enhanced plant root affinity for NO_3_^−^. Under the N0, N30, N60, and N90 conditions, the V_max_ value for the uptake of NO_3_^−^ by seedlings inoculated with Sg and Pt rose by 12.01% and 18.23%, 17.62% and 13.02%, 23.43% and 13.08%, and 27.39% and 38.07%, respectively. However, the V_max_, K_m_, and α values for NH_4_^+^ absorption were higher than those for NO_3_^−^. These results suggest that the *P. massoniana* seedlings preferentially absorbed ammonium N relative to nitrate N, while EMF inoculation enhanced the absorption of both of these forms of N (Table 4 and Table 5).

### 3.4. Root AMT Gene Expression

Transcriptional analyses revealed significant differences in the expression of four AMT genes over the different treatment conditions (Figure 4). These included three members of the AMT1 family (AMT1.1, AMT1.3, and AMT1.5) and one of the AMT family (AMT2.3). With increasing levels of N addition, the expression levels of all three of these AMT1 family genes rose in the roots of *P. massoniana* seedlings before reaching the maximal expression levels under the N60 conditions and then declining. The relative expression levels of AMT1.1, AMT1.3, and AMT1.5 were significantly higher after inoculation with EMF relative to the control conditions. Conversely, AMT2.3 expression declined with increasing N concentrations before stabilizing, and there was a significant negative correlation with AMT1.1 and AMT1.3 genes (Figure 5). But The two-way ANOVAs further confirmed that there was no correlation between AMT2.3 gene and EMF (Table 6). Under the N0 conditions, the AMT2.3 expression in plants inoculated with Sg and Pt was significantly increased by 59.37% and 96.33%, respectively, relative to the CK conditions. The three AMT1 family genes included in these analyses may represent the primary high-affinity NH_4_^+^ transporters responsible for ammonium salt uptake by *P. massoniana* roots.

### 3.5. Root NRT Gene Expression

N addition, mycorrhizal symbiosis, and interactions between N addition and EMF inoculation significantly affected the expression of the NO_3_^−^ transporter-related NRT1.5, NRT2.4, NRT3.1, and NRT3.2 genes in *P. massoniana* seedling roots. All four of these NRT genes were downregulated as the amount of added N increased. Under any given level of N addition, EMF inoculation was associated with significant reductions in the expression of all four NRT genes, with NRT1.5 being significantly downregulated under the N60 conditions and NRT2.4, NRT3.1, and NRT3.2 being significantly downregulated under the N30 conditions. No significant differences in the expression of these four genes were observed as a function of EMF inoculation with changing N concentrations after the downregulation of the four genes. At the N0 level, the expression levels of NRT2.4, NRT3.1, and NRT3.2 were significantly lower in plants inoculated with Sg and Pt relative to CK plants, with respective decreases of 70.08% and 61.16%, 40.63% and 56.32%, and 43.96% and 52.98%. NH_4_NO_3_ addition was thus able to negatively regulate the expression of all four analyzed NRT genes in *P. massoniana* seedling roots, with all of these genes serving as inducible transporters (Figure 6).

## 4. Discussion

### 4.1. Root Activity and Absorbing Area

Nitrogen is an essential nutrient that can influence plant photosynthetic activity and productivity, with plants regulating N absorption, utilization, and transformation while reducing ineffective N loss, ultimately shaping N-use efficiency. The interplay among these mechanisms can have cascading effects on plant growth and nutrient absorption [35,36]. Plant root morphological and physiological characteristics are closely related to N absorption capacity, and N is the mineral element that has the greatest impact on plant root morphology [37]. For example, when *Triticum aestivum* roots were exposed to low N conditions, increases in lateral root length and number contributed to an overall increase in the root surface area, thus enhancing N uptake by the plants. Appropriate N levels can contribute to enhanced root elongation and biomass accumulation, whereas excessively high N concentrations can adversely impact root systems by reducing overall root length, diameter, and biomass [38]. In this study, N addition was associated with increases in *P. massoniana* seedling root dry weight, activity, active absorbing area, and total absorbing area, which reached maximal levels at the intermediate N concentrations (N60) before declining, in line with previous reports on *Populus canadensis* [39]. The addition of optimal N levels can benefit root biomass accumulation and morphology, contributing to the improved uptake of N, water, and a range of other nutrients, thereby enhancing aboveground plant growth. When N levels are sufficient, developed roots can obtain higher levels of the nutrients necessary to meet their needs for growth and expansion, resulting in a wider absorption area that is better able to facilitate plant nutrient absorption [13].

Two primary pathways govern atmospheric N incorporation into the terrestrial ecosystem, namely, the deposition of atmospheric N and biological N fixation. Atmospheric N can be deposited in various forms including N_2_, NO, NO_3_^−^, and NH_3_ that must be immobilized by microbes that are present in the soil and associated with plants [40]. Mycorrhizal fungi can facilitate N transformation while providing a mechanism for host plant N storage through the mycelial absorption of inorganic soil N, which is then converted into arginine and decomposed into NH_4_^+^ and NO_3_^−^ through chitinase activity within the mycelia, followed by transport into host plants [41,42]. In symbiotic host plant root relationships, these EMF mycelia can replace root hairs and facilitate nutrient absorption. Changes in root micellization result in exogenous enzyme production and the uptake and transport of N [43]. In the present analysis, EMF inoculation was associated with increases in *P. massoniana* seedling root dry weight, root activity, root active absorbing area, and total absorbing area under different levels of N addition. When plants grow effectively, N levels are maintained within an appropriate range. Outside of this range, however, the disruption of physiological homeostasis can adversely impact plant growth, reducing plant productivity in response to N deposition [44]. EMF inoculation can improve host plant growth through the expansion of the host plant root absorbing area. When soil N levels are low, EMF can aid root N uptake through symbiotic interactions between EMF mycelia and host roots, and plants that engage in symbiotic relationships with mycorrhizal fungi exhibit higher levels of extracellular enzyme activity such that they can more readily take up and utilize N relative to non-symbiotic plants and therefore more effectively grow in low N soil environments [45]. EMF cells will not transport N to host plants until they have met their own N demands, but under high levels of N availability, increased root morphology can facilitate improved N absorption to support mycorrhizal and mycelial growth, thereby maintaining N levels within an acceptable range [46].

### 4.2. NH_4_^+^ and NO_3_^−^ Uptake Kinetics

The uptake and transport pathways that plants use for NH_4_^+^ and NO_3_^−^ differ, contributing to important differences in the uptake kinetics for these two different forms of N in plant roots. A maximum absorption threshold limits plant NH_4_^+^ and NO_3_^−^ uptake. When environmental NH_4_^+^ and NO_3_^−^ levels are below 1 mM, the uptake of these minerals is governed by a high-affinity transport system with an absorption curve that conforms to Michaelis–Menten kinetics [47,48]. Indeed, in this study, the absorption of NH_4_^+^ and NO_3_^−^ by *P. massoniana* seedlings was consistent with Michaelis–Menten kinetics. Specifically, these absorption rates rose with increasing levels of these ions in the 0.01–1 mM range and declined with increasing N concentrations. NH_4_^+^ absorption rates were higher than those for NO_3_^−^ at each tested concentration level due to the fact that NH_4_^+^ can significantly inhibit root NO_3_^−^ uptake by reducing the expression of genes encoding NO_3_^−^ cell surface transport proteins. The present data were consistent with such a model. In addition, NH_4_^+^ may alter cell membrane structural characteristics [49]. Here, N addition was associated with reductions in V_max_ values for NH_4_^+^ and NO_3_^−^. This may have been due to increases in N soil availability with higher levels of N deposition such that the plants ultimately reached maximal ion absorption threshold values, thereby reducing the V_max_ for NH_4_^+^ and NO_3_^−^ uptake. Moreover, N addition could increase the root dry weight, root activity, root active absorbing area, and total absorbing area such that the total level of absorption could increase despite a drop in the measured absorption rate [13,50].

EMF are important regulators of ecological carbon and N cycling. Relative to other microbes, EMF can more readily obtain photosynthetic products and enhance N absorption [25]. Here, EMF inoculation increased seedling NH_4_^+^ and NO_3_^−^ absorption rates such that Sg and Pt inoculation was associated with increased kinetic parameters at any given N level. The α value reflects the rate of nutrient ion influx into plant roots, with more rapid ion influx at higher α values. Sg and Pt inoculation resulted in significantly elevated α values, consistent with the ability of EMF to enhance the NH_4_^+^ and NO_3_^−^ uptake of *P. massoniana* under conditions of N deposition because inoculated plants exhibit significantly improved root morphological indices relative to non-inoculated plants. EMF inoculation is capable of improving root inorganic N absorption and utilization while also mineralizing soil organic N, transforming it into NH_4_^+^ and NO_3_^−^ that can then be absorbed and utilized by plants. The ability of plants to take up NH_4_^+^ and NO_3_^−^ is also governed by the expression of HATS and LATS genes, with different conditions and genotypes contributing to varying levels of expression for particular ammonium salt and nitrate transporter genes, thereby differentially impacting plant NH_4_^+^ and NO_3_^−^ uptake [38]. EMF are capable of regulating the expression of these transporter genes in plant roots. In addition to direct absorption pathways, plants engaged in symbiotic relationships with EMF also engage the mycorrhizal absorption pathway, and epitaxial hyphae can also absorb N and transport it to host plants, thus benefitting NH_4_^+^ and NO_3_^−^ absorption capacity [25].

### 4.3. NH_4_^+^ and NO_3_^−^ Transporter Gene Expression

Ammonium transporter gene expression patterns are governed by N levels and a range of other factors, with different AMT family genes playing environmentally appropriate roles in plant ammonium uptake. Under conditions of N stress, for example, AMT1 family genes are reportedly upregulated in *Arabidopsis thaliana* roots [51]. The majority of these genes exhibit a high level of NH_4_^+^ affinity. Subcellular localization studies have revealed that the distribution patterns of these AMT family proteins vary markedly, with AMT1.1 being primarily localized to main and lateral root epidermal and exocortical cells and to root hair acellular plasma membrane regions, thus preventing excessive NH_4_^+^ uptake by roots under high N conditions through the closure of appropriate ion channels [52]. In contrast, AMT2.3 localizes to the plasma membrane of plant cells and is responsible for controlling the NH_4_^+^ transport between the apoplast and symplast compartments [53]. Under high N concentrations, the expression of these genes is inhibited in plant roots. Here, the HATS-mediated AMT1 family gene expression in seedling roots was found to increase with N addition, peaking under the N60 conditions before declining. Relative to the CK conditions, AMT1 family gene expression rose with EMF inoculation under the four tested levels of N addition, with maximal expression under the Pt+N60 treatment conditions. In contrast, AMT2 family gene expression declined with increasing N levels, suggesting that EMF inoculation can regulate NH_4_^+^ uptake [54]. Different N sources can trigger a series of different responses in *P. massoniana* roots in a concentration-dependent manner. AMT family proteins are responsible for NH_4_^+^ transport, and the observed increases in root dry weight, root activity, active absorbing area, and total absorbing area under conditions of N addition, EMF inoculation, or a combination of the two may be attributable to AMT1 family gene upregulation and an increased ammonium supply. This model is consistent with observed NH_4_^+^ uptake kinetics, and NH_4_^+^ absorption was significantly positively correlated with ammonium transporter gene expression.

NRT1 and NRT2 family members are the primary nitrate transporters in plants, with the majority of NRT1 and NRT2 family proteins functioning as low- and high-affinity transporters, respectively [55]. The high-affinity transporters NRT2.4, NRT3.1, and NRT3.2 are induced by N in *P. massoniana*, mediating NO_3_^−^ absorption under low N concentrations. However, these transporters are inhibited by NH_4_^+^ and induced by NO_3_^−^ [56]. N addition thus reduces nitrate transporter-related gene expression, with these transporters being expressed at higher levels under low N concentrations relative to high N concentrations. Here, EMF inoculation under the N0 conditions also reduced nitrate transporter-related gene expression, whereas under the N60 and N90 conditions, such EMF inoculation increased the expression of the high-affinity NRT2.4, NRT3.1, and NRT3.2 transporters. This may be attributable to the fact that under relatively high N concentrations, mycorrhizal plants obtain additional N through mycorrhizal mycelium-mediated uptake, thereby increasing the expression of NO_3_^−^ uptake-related genes in these plant roots. AMT and NRT gene expression levels in this study were also significantly negatively correlated with one another, consistent with the fact that the *P. massoniana* root NH_4_^+^ uptake rate was higher than that of NO_3_^−^.

## 5. Conclusions

Exogenous N serves as the primary source for N uptake and utilization by plants, emphasizing the need to study the mechanisms that govern exogenous N absorption, utilization, and transport. The present results revealed that N addition and EMF inoculation influenced the NH_4_^+^ and NO_3_^−^ uptake kinetics and associated transporter gene expression levels in *P. massoniana* roots. Specifically, simulated N deposition reduced NH_4_^+^ and NO_3_^−^ absorption rates in these seedlings while promoting the upregulation of AMT1 family genes and the downregulation of NRT family genes. In contrast, EMF inoculation was associated with improved N absorption, with these effects being most pronounced at an intermediate N concentration (N60). This study was designed to offer a theoretical foundation for EMF inoculation under rising levels of N deposition associated with future climate change scenarios through the regulation of key N transport-related gene expression to enhance N utilization efficiency in a manner conducive to plant growth. These data offer an evidence base for further efforts to promote exogenous N absorption by plants and to combine N and microbial fertilizers as a means of achieving more optimal outcomes.

## Figures and Tables

**Figure 1 jof-09-00065-f001:**
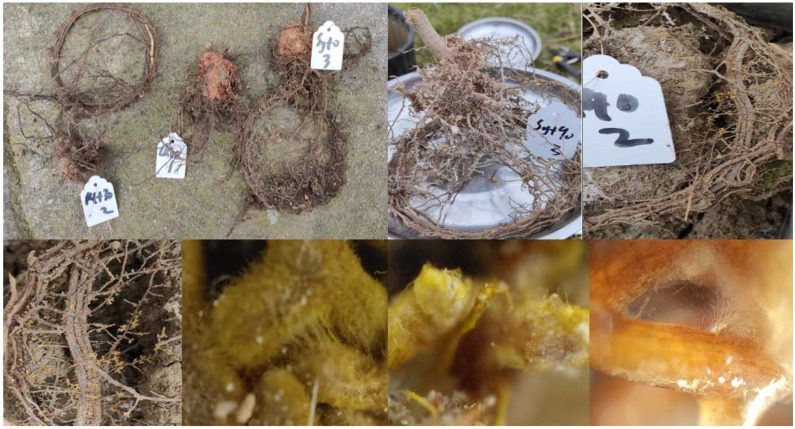
EMF formed with *Pinus massoniana* seedlings.

**Figure 2 jof-09-00065-f002:**
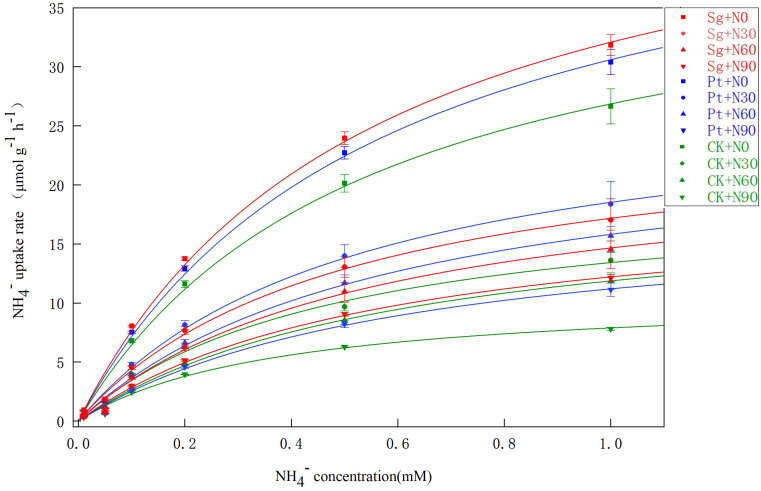
Uptake of NH_4_^+^ by *Pinus massoniana* seedlings as a function of NH_4_^+^ solution concentration. Note: Vertical bars illustrate standard errors of the means (*n* = 3). CK: control. Sg: *Suillus grevillei*. Pt: *Pisolithus tinctorius*. N0: 0 kg·N·ha^−1^a^−1^. N30: normal deposition of 30 kg·N·ha^−1^a^−1^. N60: moderate deposition of 60 kg·N·ha^−1^a^−1^. N90: severe deposition of 90 kg·N·ha^−1^a^−1^.

**Figure 3 jof-09-00065-f003:**
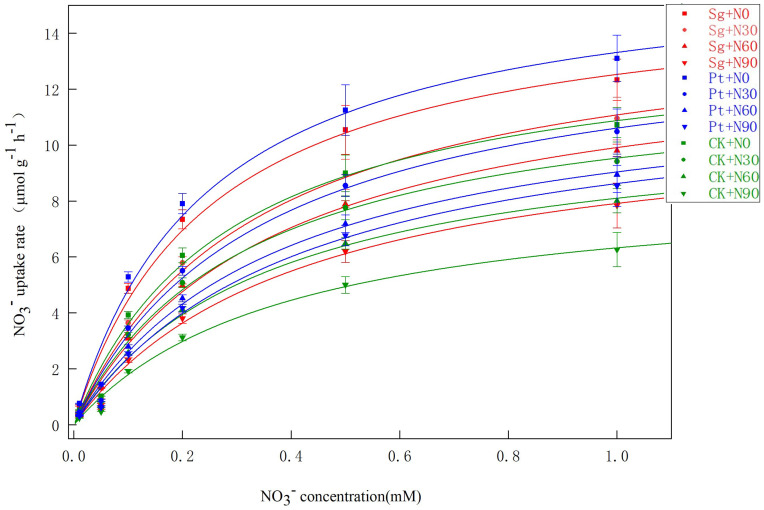
Uptake of NO_3_^−^ by *Pinus massoniana* seedlings as a function of NO_3_^−^ solution concentration. Note: Vertical bars illustrate standard errors of the means (*n* = 3). CK: control. Sg: *Suillus grevillei*. Pt: *Pisolithus tinctorius*. N0: 0 kg·N·ha^−1^a^−1^. N30: normal deposition of 30 kg·N·ha^−1^a^−1^. N60: moderate deposition of 60 kg·N·ha^−1^a^−1^. N90: severe deposition of 90 kg·N·ha^−1^a^−1^.

**Figure 4 jof-09-00065-f004:**
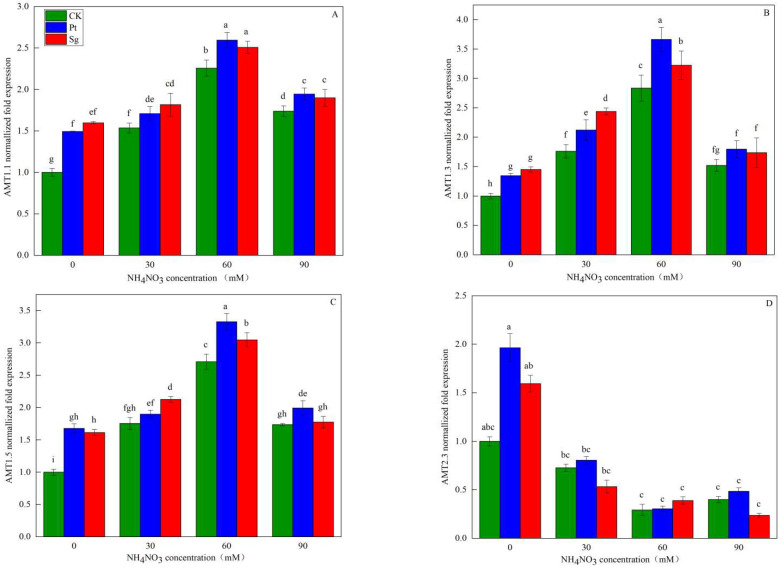
The relative expression of AMT family genes of *Pinus massoniana* seedlings as a function of NO_3_^−^ solution concentration (A, B, C and D). Note: Vertical bars illustrate standard errors of means (*n* = 3). Differences in treatments were analyzed with the Kruskal-Wallis rank sum test. Significant tests (*p* < 0.05) were followed by Duncan’s test of multiple comparisons, significant differences are indicated by lowercase letters.CK: control. Sg: *Suillus grevillei*. Pt: *Pisolithus tinctorius*. N0: 0 kg·N·ha^−1^a^−1^. N30: normal deposition of 30 kg·N·ha^−1^a^−1^. N60: moderate deposition of 60 kg·N·ha^−1^a^−1^. N90: severe deposition of 90 kg·N·ha^−1^a^−1^.

**Figure 5 jof-09-00065-f005:**
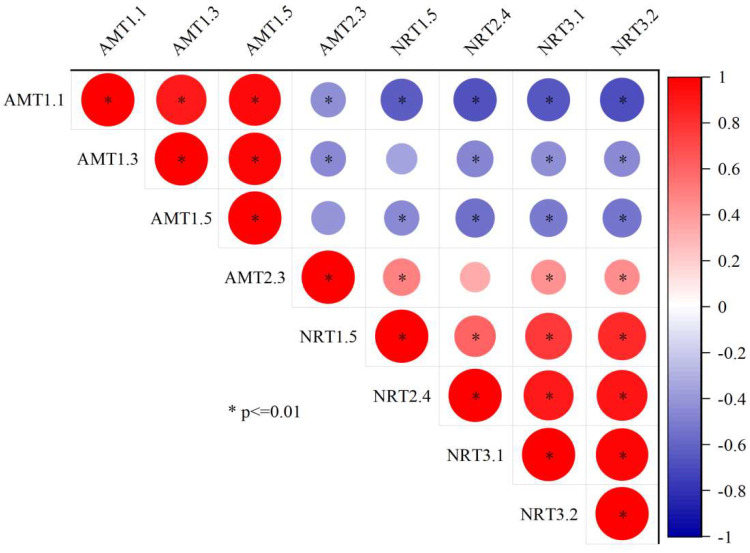
Correlation coefficients between the expression levels of AMT family genes and NRT family genes of *Pinus massoniana* seedlings. Note: Pearson correlation coefficients r were determined across all nitrogen treatments and inoculation treatments. * *p* < 0.01 (Bonferroni correction).

**Figure 6 jof-09-00065-f006:**
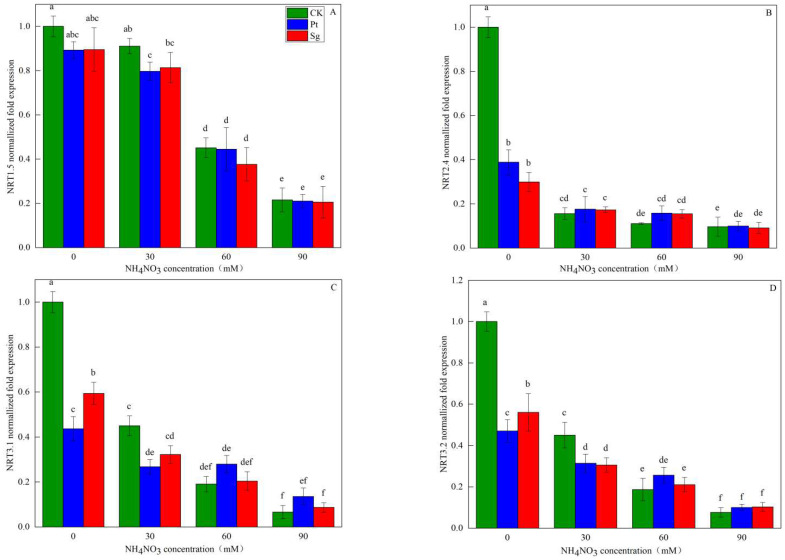
The relative expression of the NRT family genes of *Pinus massoniana* seedlings as a function of NO_3_^−^ solution concentration (A, B, C and D). Note: Vertical bars illustrate standard errors of means (*n* = 3). Differences in treatments were analyzed with the Kruskal-Wallis rank sum test. Significant tests (p < 0.05) were followed by Duncan’s test of multiple comparisons, significant differences are indicated by lowercase letters.CK: control. Sg: *Suillus grevillei*. Pt: *Pisolithus tinctorius*. N0: 0 kg·N·ha^−1^a^−1^. N30: normal deposition of 30 kg·N·ha^−1^a^−1^. N60: moderate deposition of 60 kg·N·ha^−1^a^−1^. N90: severe deposition of 90 kg·N·ha^−1^a^−1^.

**Table 1 jof-09-00065-t001:** Primers used for qRT-PCR analysis of selected genes.

Gene Name	Forward Primer	Reverse Primer
AMT1.1	TCGCTAAAGGGGAGTTTGTG	GATTATATGCGCCCCGAGTA
AMT1.2	CAGCAATCACGTCAGGTTGT	AGCTGCCAATGCGTTAAATC
AMT1.3	CCTGTTGTTGCTCATTGGCT	ATCCCACCAACCAAATGCAC
AMT2.3	GCAGCATCGTGAAGAAGAAGTGG	CCGACGGTGTAGGAGAGCATGAGCC
NRT1.5	GATCGCTTCTACTTGTTATT	TGAGCCAGTTCTTCGT
NRT2.4	GCGTTGCCTATGTCCT	TAACTGATTTCGGCTTTG
NRT3.1	TAGCCACAGAATCCTATCAA	GGGCAGAGCACCAACA
NRT3.2	CCATTGTATGCCTCTT	GCCTTGCTCTGATTTA

**Table 2 jof-09-00065-t002:** Effects of nitrogen application and inoculating ectomycorrhizal fungi on root absorbing areas of *Pinus massoniana* seedlings.

N Application (kg N ha^−1^a^−1^)	Inoculation	Root Dry Weight (g)	Root Activity (μg g^−1^ h^−1^)	Active Absorbing Area (cm^2^)	Total Absorbing Area (cm^2^)
0	Sg	4.05 ± 0.19 e	50.88 ± 4.91 f	62.51 ± 6.54 g	126.42 ± 13.26 g
	Pt	3.97 ± 0.30 e	49.85 ± 5.03 fg	58.86 ± 2.34 gh	119.87 ± 7.08 gh
	CK	3.31 ± 0.58 f	46.08 ± 4.87 g	52.51 ± 6.54 h	110.79 ± 8.59 h
30	Sg	5.23 ± 0.16 d	59.93 ± 3.97 e	81.18 ± 9.24 ef	161.10 ± 17.09 e
	Pt	5.81 ± 0.53 d	60.92 ± 4.96 de	84.54 ± 8.44 e	168.26 ± 10.11 e
	CK	4.35 ± 0.16 e	53.05 ± 6.00 f	75.42 ± 9.64 f	151.92 ± 13.92 f
60	Sg	9.83 ± 0.25 a	76.85 ± 5.35 b	126.12 ± 8.85 ab	249.01 ± 8.03 b
	Pt	10.30 ± 0.15 a	83.70 ± 6.64 a	129.90 ± 12.14 a	261.86 ± 13.66 a
	CK	8.67 ± 0.43 b	71.97 ± 4.74 bc	118.24 ± 9.19 b	238.12 ± 5.64 c
90	Sg	7.17 ± 0.68 c	70.93 ± 4.19 c	115.54 ± 12.26 bc	229.83 ± 1.93 cd
	Pt	7.48 ± 0.62 c	70.57 ± 5.09 c	116.25 ± 10.37 c	237.19 ± 19.82 cd
	CK	5.72 ± 0.34 d	65.26 ± 5.46 d	110.51 ± 1.91 d	220.61 ± 14.51 d

Note: CK: control. Sg: *Suillus grevillei*. Pt: *Pisolithus tinctorius*. N0: 0 kg·N·ha^−1^a^−1^. N30: normal deposition of 30 kg·N·ha^−1^a^−1^. N60: moderate deposition of 60 kg·N·ha^−1^a^−1^. N90: severe deposition of 90 kg·N·ha^−1^a^−1^. Numbers show the weighted mean ± standard deviation of each treatment. Standard error is shown in brackets (*p* < 0.05). Different letters indicate significant differences. (Differences in treatments were analyzed with the Kruskal–Wallis rank sum test. Significance tests (*p* < 0.05) were followed by Duncan’s test of multiple comparisons.)

**Table 3 jof-09-00065-t003:** Effects of nitrogen treatment (N), ECM treatment (ECM), and N × ECM on the parameters of *Pinus massoniana* seedlings.

Parameters	ECM		N		ECM × N	
	*F*	*p*	*F*	*p*	*F*	*p*
Root dry weight	14.98	0.00 **	130.49	0.00 **	12.99	0.00 **
Root activity	4.98	0.02 *	343.29	0.00 **	4.59	0.00 **
Active absorbing area	68.79	0.00 **	20.28	0.00 **	60.34	0.00 **
Total absorbing area	5.70	0.01 *	936.69	0.00 **	15.57	0.00 **
V_max_(NH_4_^+^)	62.86	0.00 **	24.59	0.00 **	36.22	0.00 **
K_m_(NH_4_^+^)	1.68	0.77 **	2.57	0.29 ^ns^	0.29	0.96 ^ns^
α(NH_4_^+^)	8.38	0.00 **	17.81	0.00 **	5.94	0.00 **
V_max_(NO_3_^−^)	46.52	0.00 **	16.48	0.00 **	25.37	0.00 **
K_m_(NO_3_^−^)	1.51	0.63 ^ns^	4.91	0.00 **	0.37	0.89 ^ns^
α(NO_3_^−^)	23.63	0.00 **	18.26	0.00 **	6.83	0.04 *

Note: * *p* < 0.05; ** *p* < 0.01; ns: non-significant.

**Table 4 jof-09-00065-t004:** The kinetic parameters of NH_4_^+^ absorption in the roots of *Pinus massoniana* seedlings after nitrogen application and the inoculation of ectomycorrhizal fungi.

N Application (kg N ha^−1^a^−1^)	Inoculation	V_max_ (μmol g^−1^ h^−1^)	K_m_ (mM)	α (10^−3^ mL g^−1^ h^−1^)
0	Sg	49.79 a	0.55 bc	90.52 a
	Pt	48.14 a	0.57 b	84.01 a
	CK	41.40 b	0.54 bc	76.66 b
30	Sg	25.74 cd	0.50 cd	51.60 c
	Pt	28.16 c	0.52 c	54.27 c
	CK	19.78 de	0.47 d	42.08 d
60	Sg	22.75 d	0.55 bc	41.37 d
	Pt	25.04 cd	0.33 e	75.89 b
	CK	19.22 de	0.62 a	31.09 ef
90	Sg	19.20 de	0.57 b	33.52 e
	Pt	18.09 e	0.62 a	29.41 f
	CK	10.88 f	0.37 e	29.10 f

Note: CK: control. Sg: *Suillus grevillei*. Pt: *Pisolithus tinctorius*. N0: 0 kg·N·ha^−1^a^−1^. N30: normal deposition of 30 kg·N·ha^−1^a^−1^. N60: moderate deposition of 60 kg·N·ha^−1^a^−1^. N90: severe deposition of 90 kg·N·ha^−1^a^−1^. Numbers show the weighted mean ± standard deviation of each treatment. Standard error is shown in brackets (*p <* 0.05). Different letters indicate significant differences (Differences in treatments were analyzed with the Kruskal–Wallis rank sum test. Significance tests (*p* < 0.05) were followed by Duncan’s test of multiple comparisons.)

**Table 5 jof-09-00065-t005:** The kinetic parameters of NO_3_^−^ absorption by roots of *Pinus massoniana* seedlings after nitrogen application and inoculation with ectomycorrhizal fungi.

N Application (kg N ha^−1^a^−1^)	Inoculation	V_max_ (μmol g^−1^ h^−1^)	K_m_ (mM)	α (10^−3^ mL g^−1^ h^−1^)
0	Sg	15.67 ab	0.25 e	62.44 a
	Pt	16.54 a	0.24 e	68.36 a
	CK	13.99 cd	0.29 de	48.77 b
30	Sg	14.82 b	0.34 cd	43.97 bc
	Pt	14.24 c	0.34 cd	41.65 bc
	CK	12.60 e	0.32 d	39.26 c
60	Sg	13.59 d	0.37 bc	36.64 cd
	Pt	12.45 e	0.38 bc	33.10 d
	CK	11.01 f	0.36 bcd	30.75 de
90	Sg	11.21 f	0.42 a	26.95 ef
	Pt	12.15 e	0.40 ab	29.69 e
	CK	8.80 g	0.39 abc	22.56 f

Note: CK: control. Sg: *Suillus grevillei*. Pt: *Pisolithus tinctorius*. N0: 0 kg·N·ha^−1^a^−1^. N30: normal deposition of 30 kg·N·ha^−1^a^−1^. N60: moderate deposition of 60 kg·N·ha^−1^a^−1^. N90: severe deposition of 90 kg·N·ha^−1^a^−1^. Numbers show the weighted mean ± standard deviation of each treatment. Standard error is shown in brackets (*p* < 0.05). Different letters indicate significant differences (Differences in treatments were analyzed with the Kruskal–Wallis rank sum test. Significance tests (*p* < 0.05) were followed by Duncan’s test of multiple comparisons.)

**Table 6 jof-09-00065-t006:** Effects of ECM treatment (ECM), nitrogen treatment (N), and N × ECM on the parameters of *Pinus massoniana* seedlings.

Parameters	ECM		N		ECM × N	
	*F*	*P*	*F*	*P*	*F*	*P*
AMT1.1	11.48	0.01 **	55.29	0.00 **	5.50	0.00 **
AMT1.3	77.91	0.02 *	174.19	0.00 **	21.08	0.00 **
AMT1.5	37.29	0.00 **	86.07	0.00 **	10.36	0.00 **
AMT2.3	1.359	0.32 ^ns^	14.49	0.00 **	0.583	0.74 ^ns^
NRT1.5	4.683	0.04 *	280.02	0.00 **	0.928	0.49 ^ns*^
NRT2.4	90.71	0.00 **	90.71	0.00 **	116.86	0.00 **
NRT3.1	35.61	0.00 **	276.64	0.01 *	30.83	0.00 **
NRT3.2	12.43	0.00 **	92.52	0.00 **	11.89	0.00 **

Note: * *p* < 0.05; ** *p* < 0.01; ns: non-significant.

## Data Availability

The data presented in this study are available on request from the corresponding authors. The data are not publicly available due to privacy.

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
