# Peer review of "The Relationship between Ectomycorrhizal Fungi, Nitrogen Deposition, and Pinus massoniana Seedling Nitrogen Transporter Gene Expression and Nitrogen Uptake Kinetics"

_jof, 2022, doi:10.3390/jof9010065_

Round 1

Reviewer 1 Report

JoF The Relationship between Ectomycorrhizal Fungi, Nitrogen Deposition, and Pinus massoniana Seedling Nitrogen Trans-porter Gene Expression and Nitrogen Uptake Kinetics

General comments:

It is good piece ecological work but the names of EMF fungi which are more than 25000 should be mentioned, it is being recommended with major corrections. English needs some revisions.

 Abstract:

Line 16,  a field experiment

Line 20, word increases should be replace with increase

Line 35 pinus should be italicized

 Key words:

Key words should be revise as title mentioned word should not be repeated in key words generally

Introduction:

 English need some revisions. Sentence making can be improved

N should be replace with N2

Introduction was started as the N is a biggest problem of the world even it 78% in our natural environment, it should be started optimistically by introducing EMF and their ecological role.

Reference 1,2 are also not justifying the statement, these are referred for?

Line 92, EMF plays is wrong EMF play

Introduction need severe revisions, it should be started optimistically from the significance of EMF and their positive roles and then anything can be written sequence wise.

It is looking good piece of work but o have two main questions:

1,  How the EMF were identified from inoculated roots, should be re confirmed either these roots inoculated with EMF. Simply by using ITSF and ITS4 marker primer pair, can be identified. Secondly, these EMF fungi were identified already, the herbarium voucher numbers should be mentioned.

2. How the expression of different gene groups was determined by quantifying the proteins expressed in each plant group or by any other method

Reviewer 2 Report

I have read the article under the titleThe Relationship between Ectomycorrhizal Fungi, Nitrogen
Deposition, and
Pinus massoniana Seedling Nitrogen Transporter Gene Expression and Nitrogen Uptake Kinetics” with great interest.  The paper aimed to inspect the effects of different ectomycorrhizal fungi under different nitrogen supplementation on root activity, root absorbing area, the expression of ammonium and nitrate transporter genes as well as the kinetics of these ions in Pinus massoniana seedlings.

The MS is well-written and well-conducted but there are some critical issues that should be considered to improve the quality of the manuscript. The main questions are as follows:

The abstract lacks a full explanation of the experimental design. Likewise in line 26, it was not previously explained what N60 and N90 abbreviations stand for, and which additions of nitrogen were tested.

The introduction is well-written, although small grammar changes need to be done.

Line 50. ammonium ions.

Line 71. the word salt should be replaced by ion

Line 109. Could symbiosis, should be: could form a symbiosis

The species name Pinus massoniana should be italic throughout the whole MS.

The font size needs to be unified throughout the text.

The major gaps are present in the MM section.

There is no information about the sampling. How did the authors form a representative sample from 100 seedlings? How many repetitions were taken for each analysis, especially for qPCR analysis?

Which reference genes were used in the qPCR analysis?

Did the authors test the primers’ responsiveness? How did they synthesize the primers or took the sequence from previous works and which ones?

Was the time of 3 months enough for the ectomycorrhizal development?

How authors checked and confirmed if the inoculation with different ectomycorrhizal was successful?

It was not well explained what CK stands for, and how the authors proved that in CK as control no other mycorrhizal fungi were present. Could authors thoroughly explain the program that they used for soil sterilization (line 146)?

May authors provide a detailed process of ectomycorrhizal inoculation in the MM section?

Did the authors do any nitrogen analysis after the supplementation to confirm that the supplementation treatments were well established?

May authors explain more thoroughly the principles of the wet chemistry analyzer Smartchem200 instrument?

Can authors emphasize and outline also, in the conclusion section, the difference in the effects of applied ectomycorrhizal fungi Suillus grevillei (Sg) and Pisolithus tinctorius (Pt) upon all inspected parameters such as nitrate and ammonium ion kinetics, and transporter gene expression? Which ectomycorrhizal fungi had a higher influence on nitrogen metabolism and in which way precisely?

Round 2

Reviewer 1 Report

Dear Colleague,

Hoping you are fine. now it is in better form than previous version with one revision that is required.

 The pictures of ectomycorrhizae/morphotypes of inoculated roots with two EMF should be made part of this paper in the form of figure.

 Remaining is ok from my side.